# Triazole-Functionalized Mesoporous Materials Based on Poly(styrene-*block*-lactic acid): A Morphology Study of Thin Films

**DOI:** 10.3390/polym14112231

**Published:** 2022-05-31

**Authors:** Melisa Trejo-Maldonado, Aisha Womiloju, Steffi Stumpf, Stephanie Hoeppener, Ulrich S. Schubert, Luis E. Elizalde, Carlos Guerrero-Sanchez

**Affiliations:** 1Centro de Investigación en Química Aplicada, Bldv. Enrique Reyna, No. 140, Saltillo 25100, Mexico; melisa.trejo.d18@ciqa.edu.mx; 2Laboratory of Organic and Macromolecular Chemistry (IOMC), Friedrich Schiller University Jena, Humboldtstr. 10, 07743 Jena, Germany; aisha.womiloju@uni-jena.de (A.W.); steffi.stumpf@uni-jena.de (S.S.); s.hoeppener@uni-jena.de (S.H.); ulrich.schubert@uni-jena.de (U.S.S.); 3Jena Center for Soft Matter (JCSM), Friedrich Schiller University Jena, Philosophenweg 7, 07743 Jena, Germany

**Keywords:** functional block copolymers, triazolyl, thin films, solvent vapor annealing, porous membranes

## Abstract

We report the synthesis of poly(styrene-*block*-lactic acid) (PS-*b*-PLA) copolymers with triazole rings as a junction between blocks. These materials were prepared via a ‘click’ strategy which involved the reaction between azide-terminated poly(styrene) (PS-N_3_) and acetylene-terminated poly(D,L-lactic acid) (PLA-Ac), accomplished by copper-catalyzed azide-alkyne cycloaddition reaction. This synthetic approach has demonstrated to be effective to obtain specific copolymer structures with targeted self-assembly properties. We observed the self-assembly behavior of the PS-*b*-PLA thin films as induced by solvent vapor annealing (SVA), thermal annealing (TA), and hydrolysis of the as-spun substrates and monitored their morphological changes by means of different microscopic techniques. Self-assembly via SVA and TA proved to be strongly dependent on the pretreatment of the substrates. Microphase segregation of the untreated films yielded a pore size of 125 nm after a 45-min SVA. After selectively removing the PLA microdomains, the as-spun substrates exhibited the formation of pores on the surface, which can be a good alternative to form an ordered pattern of triazole functionalized porous PS at the mesoscale. Finally, as revealed by scanning electron microscopy–energy dispersive X-ray spectroscopy, the obtained triazole-functionalized PS-porous film exhibited some affinity to copper (Cu) in solution. These materials are suitable candidates to further study its metal-caption properties.

## 1. Introduction

The preparation of block [1], spontaneous gradient [2], gradient or asymmetric [3,4] copolymers remains in the spotlight due to the intrinsic self-assembly properties of these macromolecules. Furthermore, the synthesis of functional polymers is an additional area of interest due to the numerous possibilities that arise from adding and transforming molecules with different functionalities by exploring different synthetic pathways. Specifically, block copolymers are sought-after due to their capability to segregate into periodically ordered structures.

Block copolymers are macromolecules that contain in their structure at least two different blocks, which are often of dissimilar chemical nature. The multivalence between blocks provides these systems with singular physicochemical properties while being covalently bonded [5]. A common example of these systems is amphiphilic diblock copolymers, in which the difference in the polarity between blocks promotes a self-assembly process of the phases to obtain ordered structures at the nanoscale [6]. Such is the case of the poly(styrene-*block*-lactic acid) (PS-*b*-PLA) copolymer system, which has been widely studied due to its phase segregation properties. To observe and gain a better understanding of the phase segregation in these block copolymers, the preparation of thin films has become a resourceful tool to study their self-assembly process under mild experimental conditions [7,8]. Thin film nanopatterns, commonly observed with microscopic techniques such as scanning electron microscopy (SEM), can yield specific morphologies according to the chemical nature and the volume fraction of the blocks [9]. PS-*b*-PLA-based copolymers are also utilized as templates to prepare nanopatterns in which the PLA block is selectively removed to ultimately obtain a PS-porous membrane [10].

Furthermore, it is highly pursued to incorporate functional groups at a certain stage during the preparation of porous membranes due to the potential applications that these moieties can provide to the final nanostructure. In this regard, Poupart et al. [11] reported the preparation of functional porous structures based on PS-*b*-PLA copolymer templates by synthesizing an aldehyde group that remains after removal of the PLA minor block. Other reports have also investigated the thin film behavior of diblock copolymers considering the presence of functional groups. Piñón et al. [12], studied the PMMA-*b*-PS copolymer system by performing a sulfonation reaction of the PS block during the synthetic pathway.

One of the highlights of the preparation of porous membranes through the self-assembly of a block copolymer is the possibility of modifying their chemical structure throughout synthesis. A controlled environment on the polymerization of the precursors combined with functionalization reactions can yield stable macromolecules. In addition, the incorporation of ‘click’ chemistry via cupper-catalyzed reactions allows targeting a specific molecule due to its stereo-selective mechanism [13]. It is noteworthy that, after etching one of the nano-template phases, functional groups embedded onto porous matrices are highly likely to preserve their chemical properties due to the existing covalent bond [14]. Thus, incorporating functional molecules that can modify the properties of block copolymers remains a widely unexplored field as practically unlimited possibilities arise; for instance, from combining reversible deactivation radical polymerization (RDRP) techniques and classic organic chemistry reactions.

In this context, we have previously reported an analogous system that considers the preparation of triazole-functionalized porous polystyrene by incorporating a functional comonomer (i.e., 4-azidomethylstyrene) into the PS block. The final porous matrix exhibited a hint of an ordered structure on the morphology of a hollowed surface [15]. Thus, in the present contribution, triazole derivatives were used to link two homopolymer blocks. We hypothesize that a minimum ratio of triazole groups would allow the biphasic system to self-assemble without drastically modifying the polarity of the PS block. Hence, we assumed that phase segregation of this block copolymer system would not be largely affected by the presence of the triazole derivative. Once the copolymer segregated, the PLA block was selectively removed to obtain an ordered porous matrix. With this approach, the triazole moieties would still be present to provide functional properties to the final porous PS matrix.

It is worth mentioning that triazole derivatives have binding capabilities towards metallic centers, such as Cu^2+^ or Zn^2+^ [16]. In principle, this feature would allow the membrane to capture such metallic species through a coordination bond between triazole rings and cations. This type of membrane is used for numerous applications, such as water purification [17], antifungal and antimicrobial agents [18,19], CO_2_ adsorption [20], drug encapsulation and release [21], etc.

It has been well established that block copolymers can be synthesized by RDRP techniques in a controlled environment. In this contribution, polystyrene and polylactic acid were synthesized by Activators ReGenerated by Electron Transfer–Atomic Transfer Radical Polymerization (ARGET–ATRP) and Ring Opening Polymerization (ROP), respectively, to be used as precursors in a subsequent ‘click’ chemistry coupling reaction. Hence, in this work, we address the synthesis of triazole-functionalized PS_n_-*b*-PLA_m_ via a Cu-catalyzed click reaction of an azide-terminated polystyrene (PS-N_3_) with an acetylene-terminated poly(lactic acid) (PLA-Ac). Afterwards, we focused on the preparation of thin films based on one of the synthesized functional copolymers. We approached this by performing solvent vapor annealing (SVA), thermal annealing (TA), and hydrolysis of the as-spun films and by monitoring morphological changes via scanning electron microscopy (SEM) and atomic force microscopy (AFM). Thus, our approach also evaluated the potential effect of the triazole functionality on the overall behavior and morphology of the self-assembled block copolymer thin films and porous membranes derived thereof. Finally, we examined the presence of metal moieties of the obtained porous materials by scanning electron microscopy-energy dispersive X-ray (SEM-EDX) analysis.

## 2. Materials and Methods

### 2.1. Materials

Tetrahydrofuran (THF), dichloromethane (CH_2_Cl_2_), methanol (CH_3_OH), and *N*,*N*-dimethylformamide (DMF) were dried using the solvent purification system PureSolv—EN™ (Innovative Technology, Stratham, NH, USA). Toluene (anhydrous), ethanol, acetone, and *n*-hexane were obtained from commercial suppliers and used without any further purification procedures unless otherwise stated. Copper (I) bromide (Aldrich, Saint Louis, MO, USA, 99%) was purified by stirring in glacial acetic acid; thereafter, it was thoroughly washed with methanol and anhydrous diethyl ether and dried under vacuum overnight [22]. *N*,*N*,*N*,*N*″,*N*″-Pentamethyl diethylenetriamine (PMDETA, 99%) and hexamethyldisilazane (HMDS) were obtained from Aldrich (Saint Louis, MO, USA) and used as received.

The synthesis of the polymeric precursors—azide terminated polystyrene, PS-N_3_; and acetylene terminated polylactic acid, PLA-Ac—is described in the Appendix A. Purification of the synthesized copolymers was performed in a chromatographic column using aluminum oxide (neutral, 0.063–0.200 mm, Merck, Saint Louis, MO, USA) as stationary phase.

### 2.2. Instrumentation

^1^H nuclear magnetic resonance (NMR) spectra of the obtained copolymers were measured on a 400 MHz Bruker Avance III spectrometer (Bruker Corporation, Karlsruhe, Germany) in deuterated chloroform (CDCl_3_) at room temperature using 128 scans with 2 s delay between scans and tetramethylsilane (TMS) as an internal standard.

The molar mass of the copolymers was determined using a size exclusion chromatography (SEC) Shimadzu system with a mixture of chloroform, triethylamine, and 2-propanol (94:4:2 *v*/*v*/*v*) as eluent at a flow rate of 1 mL min^−1^. The system is equipped with an CBM-20A controller, a LC-10AD pump, a RID-10A refractive index detector, and a PSS SDV linear S, 5 μm, column (Polymer Standards Service, PSS, Mainz, Germany). The average number molar mass (*M*_n_) was estimated against a calibration curve built with polystyrene (PS) standards of narrow dispersity (*Ð*).

Scanning electron microscopy (SEM) imaging was performed using a Zeiss Sigma VP field emission scanning microscope equipped with an Everhart–Thornley SE and InLens detector (Carl Zeiss AG, Oberkochen, Germany), using an accelerating voltage of 10 kV. The samples were coated with a thin layer of platinum via sputter coating (CCU-010 HV, Safematic, Zizers, Switzerland) prior the measurement. Block copolymer films were also investigated under ambient condition by means of atomic force microscopy (AFM) NTegra Aura (NT-MDT) in tapping mode utilizing cantilevers (NSC35/AlBS, MicroMasch, Tallinn, Estonia) under hard tapping conditions, i.e., high ratio of free amplitude of cantilever oscillation and set point. 

### 2.3. Synthesis of PSn-b-PLAm Diblock Copolymers (‘Click’ Chemistry Coupling Reaction)

2.0 mg (0.0138 mmol) of CuBr(I), 234.0 mg (0.0138 mmol) of PS-N_3_ and 287.0 mg (0.0277 mmol) of PLA-Ac were weighed in a round bottom flask and immediately connected to a stream of nitrogen gas. Next, 5–7 mL of DMF were added under stirring until full dissolution of both homopolymers. Upon dissolution, 3.0 µL (0.0138 mmol) of PMDTA were added using a degassed micropipette and the entire system was closed under an inert atmosphere. The mixture was stirred at room temperature for 24 h. Thereafter, Cu^I/^PMDETA catalyst was removed through a short column of alumina using CH_2_Cl_2_ as eluent. The final product was isolated by precipitation into an excess volume of a solvent mixture of hexane:ethanol:methanol (6:3:1, *v*/*v*/*v*). A light yellow-ish powder was isolated after drying under vacuum overnight at 40 °C.

### 2.4. Preparation of PS-b-PLA Based Thin Films

#### 2.4.1. Substrate Preparation

Silicon wafers were employed as substrates to prepare the copolymer-based thin films. In the indicated experiments, the surface of the silicon wafers was pretreated by a silylation procedure to avoid delamination of the film. The native layer of the silicon substrates was etched by dipping pieces of wafers (~1 cm^2^) into a *piranha* solution (H_2_O_2_:H_2_SO_4_, 1:3, *v*/*v*) at 100 °C for 60 min. Afterwards, the pieces were extensively rinsed with water and ultrasonicated in an acetone:methanol (1:1, *v*/*v*) mixture. After drying under a nitrogen stream, substrates were dipped into an HMDS:toluene (1:5, *v*/*v*) solution overnight. Finally, the substrates were rinsed with toluene and dried in an oven at 80 °C prior to use [23]. Substrates not treated with silylation were simply rinsed with CH_2_Cl_2_ and ethanol and further cleaned in a plasma oven for 2 min (160 cm^3^ min^−1^ of argon flow rate).

#### 2.4.2. Annealing Experiments

A 2.0% *w*/*v* PS-*b*-PLA copolymer solution was prepared in THF at room temperature. The copolymer was stirred in the solvent until full dissolution. Once dissolved, the solution was filtered using a 0.2 µm PTFE membrane to diminish the presence of contaminants. Then, 50.0 µL of the solution were spin coated onto a silicon wafer. Each substrate was spin coated for 60 s at 2000 rpm and air-dried prior to annealing.

For the solvent vapor annealing (SVA) experiments, as-spun substrates were placed inside a closed glass chamber next to an open vial filled with *o*-xylene (~5.0 mL). On the basis that *o*-xylene and toluene have similar polarity, and added to the affinity between PS and toluene, o-xylene was chosen as a selective solvent to promote the microphase segregation of the PS-domains [24]. SVA was performed at different times. After annealing, the surface was exposed to open air and allowed to evaporate any remaining solvent for a couple of minutes inside a fume hood. 

In the case of thermal annealing (TA) experiments, as-spun substrates were placed inside an oven at 120 or 150 °C for 10 min. Finally, the substrates were carefully removed and cooled to room temperature [25].

#### 2.4.3. Hydrolysis Experiments

To selectively remove the PLA domain of the films, the annealed and/or as-spun substrates were dipped into an alkaline solution (NaOH 0.5 M, H_2_O:CH_3_OH, 6:4) at different times: 5, 15, 30, and 60 min. Afterwards, each substrate was carefully extracted from the solution and rinsed with an excess of water. The substrates were dried under vacuum overnight prior to SEM analysis.

### 2.5. Preparation of PS-b-PLA Based Monoliths

#### 2.5.1. Method 1

A determined amount of a PS_n_-*b*-PLA_m_ copolymer was dissolved in toluene to yield a final concentration of 10.0% *w*/*w*. The copolymer was solubilized for several hours under continuous stirring at room temperature. Afterwards, the solution was filtered with a 0.2 µm nylon membrane into a new vial to remove any contaminants. If the solution was too viscous, the filtration step was not carried out. Thereafter, the solution was poured into an as-made Teflon mold and the solvent was allowed to evaporate for several days under a fume hood. To completely eliminate the solvent, the films were dried in a vacuum oven for 2 h at 40 °C. To promote the orientation of the domains on the pre-formed film, an annealing procedure was performed by placing the sample in an oven at 160 °C for 1 h [26]. Finally, the sample was cooled to room temperature and analyzed by SEM.

#### 2.5.2. Method 2

200 to 350 mg of a PS_n_-*b*-PLA_m_ copolymer were placed inside a PEEK (polyether ether ketone) circular mold (D = 1.5 cm). The mold was coated with a thin foil of FEP (fluorinated ethylene propylene) to avoid the copolymer from sticking to the mold and facilitate the collection of the resulting sample. The powder was evenly distributed inside the mold and then the mold was placed inside an oven at 160 °C. After 1 h, the mold was capped with a stainless-steel press of the same diameter of the mold to properly seal it. The mold was carefully placed back inside an oven at 160 °C for 1 h. Finally, the sample was allowed to cool down to room temperature and removed from the mold. 

### 2.6. Metal Uptake Experiments

A 1.0 % *w*/*w* CuCl_2_ solution was prepared in a mixture of ethanol:water (1:1, *v*/*v*). The hydrolyzed substrates were immersed in 10.0 mL of the prepared solution overnight. Afterwards, substrates were removed from the solution and thoroughly rinsed with water and methanol. Finally, the samples were freeze-dried prior to SEM analysis.

## 3. Results and Discussion

We can find in many examples in the literature that PS_n_-*b*-PLA_m_ copolymers are capable of self-assembling to form ordered structures in thin films at a nanoscale [23,27,28,29]. Our investigation aimed to prepare this type of block copolymers via a ‘click’ chemistry coupling of PS and PLA homopolymers. In the following sections, we address the results of this synthetic route and analyze the self-assembly behavior of the obtained materials.

### 3.1. Synthesis and Characterization of PS-b-PLA Block Copolymers

The synthesis of the PS_n_-*b*-PLA_m_ copolymers was executed by copper-catalyzed azide-alkyne cycloaddition (CuAAC) of the azide and acetylene end-groups of the PS-N_3_ and PLA-Ac, respectively. This synthetic approach roughly consisted of the 1,3-cycloaddition reaction of the two functional groups: azide and alkyne, these functionalities were located at the chain-ends of PS and PLA blocks, respectively. Hence, a CuAAC of azide and alkyne groups could obtain a triazole-based covalent union between PS and PLA blocks. The CuAAC strategy to build a covalent bond between blocks has been employed successfully to obtain block copolymers such as poly(styrene-*b*-*N*-isopropylacrylamide) [30]. Note that both homopolymers were separately synthesized and purified. PS-N_3_ was obtained via ARGET ATRP and followed by a substitution of the bromine end group using sodium azide. PLA-Ac was obtained from a ROP of the corresponding monomer. The experimental details of these polymerizations can be found in the Appendix A. A schematic representation of the synthesis of the block copolymers is displayed in Figure 1.

The isolated copolymers were characterized by ^1^H NMR and SEC. Table 1 summarizes the molecular features of the isolated copolymers, **1A** and **2A** contain an aliphatic triazole derivative in their structure, while compounds **1B**–**3B** contain the aromatic triazole functional group.

Average number molar mass (*M*_n_) values were estimated from the analysis of ^1^H NMR spectra of each copolymer. After obtaining the values of *n* and *m* for each block, the volume fraction (*ƒ*_A_) was calculated from the *ƒ_PLA_* equation (Table 1). *ƒ*_A_ value of the minority block is a crucial parameter to control the morphology of the film surface for the self-assembling process of the block copolymers. Theoretically, by varying the *ƒ*_PLA_ parameter in the range from 0.3 to 0.4, we could aim to obtain a cylindric morphology [31]. As gathered from Table 1, the obtained *ƒ*_PLA_ value of copolymers **1A**–**2B** is within this range, whereas the *ƒ*_PLA_ value of compound **3B** is closer to 0.5, which could lead to a gyroid or a lamellar morphology. Thus, we decided to investigate the self-assembly behavior in thin films by using copolymer **1A**, as described in following section of this manuscript.

*M*_n_, and dispersity (*Ɖ*) values of the copolymers were estimated by SEC measurements as well. Most of the copolymers exhibited a narrow molar mass distribution (*Ɖ* = 1.12–1.24) suggesting that the controlled homopolymerization of each block in combination with the click coupling reaction between both end-functionalized PS and PLA homopolymers was successful. 

By observing the *M*_n_ values reported in Table 1, it should be noted that, in most cases, these estimated values have minor variations as obtained from ^1^H NMR and SEC characterization techniques. This effect could be attributed to the size of the copolymer in solution as compared to the size of the PS standard calibration used to estimate *M*_n_ values of the copolymer. Hence, variations in the *M*_n_ values of the copolymers are expected when observing SEC results due to the chemical difference between the analyte and the standard reference [32]. It is worth mentioning that higher molar masses (*M*_n_ > 20 kg mol^−1^) promote the phase segregation in block copolymers. At lower values, the incompatibility between the blocks is weaker, which inhibits microphase segregation [33]. Thus, we aimed to obtain block copolymers with a *M*_n_ value between 20 and 30 kg mol^−1^ in order to promote phase segregation in the later stages of this investigation.

Further NMR analysis confirmed the successful outcome of the click reaction. Figure 1 displays the ^1^H NMR spectrum of PS_77_-*b*-PLA_81_, **1B**, where the most representative signals have been assigned. Copolymer **1B**, of a lower molar mass, was used to better observe the end group units of the copolymer. The broad signal within the chemical shift (δ) range from 7.24 to 6.25 ppm was assigned to the aromatic protons of the PS repetitive unit. Similarly, a broad signal at δ = 5.47–4.90 ppm was assigned to the –CH group of the PLA repetitive unit. To further confirm the presence of the triazole derivative, we proposed the following assignations to the phenyl-triazolyl junction group: δ = 8.21–8.05 ppm and 7.91–7.73 ppm corresponding to protons *F* and *G* as indicated in Figure 1. The triazolyl –CH group (proton *E*, Figure 1) was ascribed to the signal at δ = 8.03 ppm that is slightly overlapped with the aromatic protons assigned to *F*. The assigned proton signals are in good agreement with previous reports investigating similar systems. For instance, Terzic et al. reported the synthesis of poly(vinylidene fluorine) (PVDF) triblock copolymers via CuAAC [30], where the triazole proton was assigned at δ = 8.09 ppm, which is consistent with our results considering the similarity of the chemical vicinity. In addition, a diffusion-ordered spectroscopy (DOSY) NMR measurement confirmed that the copolymer was obtained as evidenced by the presence of a single diffusion coefficient signal, as it is expected for block copolymers (see Appendix A).

Note that the synthesis of our block copolymers considered two derivatives to form the triazole groups, an aliphatic and an aromatic derivative, represented by compounds A and B in Table 1, respectively. It is expected that the presence of an aromatic or aliphatic triazole derivative unit may not largely affect the phase segregation of the copolymer. However, the presence of an aromatic derivative may contribute to the complexation with a metallic center on the porous matrix due to an electronic delocalization of the aromatic rings [34].

Overall, we conclude that click coupling reaction at the investigated conditions enables modulation of the PS:PLA ratio that is required to obtain specific morphologies. At this stage, we assume that the obtained copolymers are adequate candidates to undergo microphase segregation. Thus, the physicochemical properties of PS-PLA-based copolymers will mainly enable microphase segregation regardless of the functionalization present in their structure. Hence, we expect that the triazole functionality junction in the synthesized copolymers might not interfere with the phase segregation process and remain present on the surface of the final PS-porous matrix.

### 3.2. Solvent Vapor Annealing (SVA)

As reported before, PS-*b*-PLA copolymers are eligible materials to pattern different morphologies at the mesoscale [23]. Therefore, we investigated the self-assembly of these copolymers by monitoring the morphology of prepared thin films via scanning electron microscopy (SEM). The starting copolymer solution was prepared by dissolving copolymer **1A** in THF (2.0% *w*/*w*). This solute-solvent system was chosen after comparing a first set of as-spun substrates using THF, toluene, CH_2_Cl_2_, and dioxane as solvents at the same copolymer concentration. SEM of the as-spun samples dissolved in THF exhibited a better periodicity of the formed domains on the surface as compared to the samples dissolved in toluene, CH_2_Cl_2_, or dioxane (see Appendix A). In addition, samples that were dissolved in THF also appeared to promote phase segregation by a simple spin-coating process on the silicon wafer. Hence, we decided to continue our analysis with THF-based copolymer solutions.

Regarding the annealing solvent, we selected *o*-xylene due to the polar affinity with the PS phase, which could selectively contribute to the self-assembly process on the thin film surface. On that account, several pieces of wafers were spin coated and annealed as described in the Section 2.4.2. After SVA, the substrates were air-dried inside a fume hood and analyzed by SEM.

SEM images of the as-spun substrate, shown in Figure 2A, exhibited a two-phased surface in which circle-like structures are homogeneously distributed across a continuous phase. As previously mentioned, we aimed to obtain a cylindrical morphology on the film, in which PLA domains were homogeneously distributed across the PS matrix. In this case, the obtained morphology resembles a ‘dotting’ pattern, similar to the one reported for analogous systems [8] as evidenced by the circular shapes throughout the film. Therefore, we assumed that the circular structures (0.2–1 µm) correspond to a segregated phase of PLA, since it is the minority block of the block copolymer system. Hence, the continuous phase might correspond to a PS matrix. After 15 min of SVA (Figure 2B), these structures did not exhibit a significant change compared to the morphology observed on the as-spun substrate. After 45 min of annealing with *o*-xylene (Figure 2C), the surface of the substrates exhibited a hollowed structure considering the contrast between the continuous phase and the circle-like structures. At this point, the PLA minority phase seems to be etched from the surface. Nonetheless, no attempts to remove the PLA phase were made at this stage. Thus, we hypothesize that the PLA domains formed an interphase between the PS and the silicon substrate, which may be attributed to the chemical affinity between the PLA and the silicon wafer surface. This effect was further investigated via AFM on the same substrate and subsequent image correction and analysis with Gwyddion v2.47 software. The hard tapping conditions under which the measurement was carried out allowed for the penetration of the tip into the softer phase of the block copolymer film, as revealed by the observed topography contrast between the respective domains. In this additional characterization, the holes (PLA domains)—formed after 45 min of SVA—appeared to interact with the substrate at approximately 100 nm below the continuous phase (PS) (Appendix A).

The software ImageJ was used to measure and analyze the pore-like structures of the SEM images. The mean sizes were determined with a minimum of 20 measurements on different areas of the image. According to this size analysis of the pore-like structures on the SEM image of Figure 2C, the diameter of the formed cylinders has a mean size of 125 ± 22 nm after 45 min of SVA with *o*-xylene.

Longer annealing times were performed on the rest of the spun substrates and the corresponding SEM images are shown in Figure 2D–F. In these images, a similar pattern to that of Figure 2C (45 min of SVA) can be observed. Although, after 6 h (Figure 2F), the pore homogeneity on surface of the film starts vanishing as compared to shorter annealing times. This suggests that, for longer SVA times, the solvent may begin to dissolve the film. The size analysis of the pore diameter for Figure 2D–F yielded similar values within a range of 121 to 139 nm.

After the annealing step, it is desirable to selectively remove the PLA minority phase to obtain a hollowed PS-matrix. In this regard, we attempted to remove PLA by immersing the substrates in an alkaline solution (NaOH 0.5 M). However, we observed delamination of the entire film after a couple of minutes of immersion in the alkaline medium. This effect has also been observed for analogous systems where the substrate surface is not compatible enough with the continuous phase, which leads to delamination of the film [27,35].

Consequently, we carried out a pretreatment of the substrates to avoid delamination of the film. The pretreatment of the substrates consisted in the formation of an interlayer between the block copolymer and the wafer surface. It was hypothesized that this interlayer might promote compatibility between the substrate surface and the copolymer film. As described in the materials and methods section, the surface of the silicon wafer was etched with a *piranha* solution followed by a silylation of the surface. As a result, a hydrophobic interlayer is formed between polymer film and substrate [36]. Thereafter, substrates were pretreated as indicated above and, subsequently, spin coated using the same conditions to those applied to the untreated substrates. Previous results suggested that 45 min of SVA with *o*-xylene might form cylindrical-like cavities; therefore, we attempted to replicate this procedure on pretreated substrates.

Figure 3 shows a comparison between a spin-coated sample on an untreated surface (Figure 3A) and a spin-coated sample on a pretreated surface (Figure 3B) after 45 min of SVA. The previously tested annealing conditions were not reproducible for the HMDS-treated surfaces. A noticeable difference can be observed on the pretreated surface where large hollows (1–3 µm) were formed across the surface of the film as well as some smaller pore-like structures surrounding those larger structures. Thereafter, the pores and the pattern regularity on the film were lost after annealing. Thus, we hypothesized that longer SVA times would allow the phases to reach an equilibrium into a more ordered state.

Considering this, an additional SVA experiment was performed at considerably longer annealing times: 2, 4, 8, and 16 h. Unexpectedly, longer SVA times did not promote phase segregation and the formation of morphology on the film. Based on the SEM analysis of the samples, we theorized that at longer annealing times (t > 45 min) the films seem to be prone to dewetting of the surface leading to their damage (see Appendix A for more details). The interlayer formation derived from the HMDS pretreatment had a significant influence on the self-assembly process of the investigated block copolymer system on the surface. Thus, in this case, the utilized annealing solvent and conditions did not promote the cylindrical morphology observed on the untreated substrates.

Despite the aforementioned, it should not be discarded that as-spun substrates on untreated substrates exhibited a homogeneous pattern. In this regard, we considered that the morphology of as-spun films is comparable to those observed in previous reports with analogous systems [29]. Therefore, we decided to further investigate and exploit the characteristics of as-spun films on untreated substrates and without SVA, which is discussed later on in Section 3.4. Considering these results, we explored other alternative methods to obtain ordered-porous structures, as discussed next.

### 3.3. Thermal Annealing

Thermal transitions of block copolymers have also been used to anneal microphases to access certain morphologies. To further investigate the microphase segregation behavior of copolymer **1A**, thermal annealing (TA) experiments of thin films were additionally performed. In this regard, the thermal transitions of the synthesized block copolymers were previously evaluated using differential scanning calorimetry (DSC) and thermogravimetric analysis (TGA) (Appendix A).

Regarding DSC investigations, the glass transition temperature (T_g_) of each block allowed us to determine a range of values where the copolymer is in a disordered state. Furthermore, based on TGA measurements, we can determine at which temperature the copolymer begins to decompose (T_D_). By analyzing these two parameters we proposed a range of temperature values in which the copolymer transitions may drive the system from a fluid disordered state into an ordered state. Hence, the order–disorder transition (T_ODT_) can be found above the values of the T_g_’s of both individual blocks and below the T_D_. According to the DSC and TGA measurements of the synthesized block copolymers, the proposed temperature range is 110–210 °C. Previous investigations with analogue systems report a T_ODT_ = 138 °C [33].

As discussed for previous annealing experiments, we observed delamination of the films deposited on the untreated substrates. In an attempt to prevent this effect, we also performed the silylation treatment on the wafers for the thermal annealing experiments. Considering the thermal profile of the investigated block copolymer system, we performed a thermal annealing experiment using the same procedure for the preparation of thin films as described above for the SVA experiments. Figure 4 shows SEM images of thin films thermally annealed at 120 and 150 °C for 10 min. At 120 °C (Figure 4A) the film shows an irregular distribution of PLA domains suggesting that the system did not reach an equilibrium leading to a cylindrical phase segregation at the investigated conditions. Two size populations of the pore-like structures are visible in this image, where the smaller pores seem to merge with each other to form larger structures with an irregular shape. Although, at this point, this latter statement is only a hypothesis of what may be occurring.

At 150 °C (Figure 4B) the film presents a noticeable difference between the size populations of PLA domains. It has been reported that higher temperatures promote the formation of more uniform patterns in PS-PLA based thin films [36]. However, this was not the case for the observed films, as there is a greater gap between the size populations of the formed structure and, somewhat, a more irregular pattern after increasing the temperature to 150 °C.

The thermally annealed films exhibited a different morphology pattern compared to that obtained with the SVA method on treated substrates. At a first glance, TA might seem to promote a more regular arrangement of the domains regarding the size of the pore-like structures as opposed to the observed with SVA pretreated sample, where the difference between the size of larger and smaller pores was more notable (Figure 3B). However, the pore-like structures in the thermally annealed films yielded two size populations, which might lead to irregularities in the resulting film once the PLA phase is selectively removed.

Thermal annealing of the thin films may represent an alternative to promote orientation of the PS-PLA domains by varying some parameters. In other reports [36], the use of a vacuum oven, instead of a conventional oven, at a higher temperature can provide thin films with fewer irregularities as the probability of PLA decomposition is reduced due to the presence of less oxygen and/or humidity.

### 3.4. Hydrolysis

At this stage, we would like to outline some of the findings: (a) SVA of untreated surfaces yielded an acceptable homogeneity on the film surface; however, it led to delamination when attempting to selectively remove the PLA domains; (b) SVA of the pretreated substrates did not exhibit the same homogeneity and regularity on the morphology of the film as it did for untreated substrates; (c) TA of the pretreated substrates under the experimental conditions did not promote the formation of the desired morphology of the film; (d) spin coating of the substrates seemed to promote the segregation of the phases towards a cylindric-like morphology (Figure 5A). Henceforth, this section addresses this last premise and explores the possibility of hydrolyzing the PLA of as-spun films, meaning that no annealing was performed for these substrates prior to the hydrolysis.

Following the same spin-coating process used for prior samples, an as-spun film was dipped into an alkaline solution to selectively remove the PLA domains and obtain a PS-porous film matrix. Figure 5B shows an SEM image of the hydrolyzed as-spun pretreated substrate where a difference in contrast between the formed pores and the PS continuous phase can be distinguished. By comparing the as-spun film (Figure 5A) and the corresponding hydrolyzed film (Figure 5B), we can conclude that the PLA was removed according to the difference in the density of the pores between images. The regularity and the condition of the film seem to be unaffected by the hydrolysis procedure, suggesting that the silylation pretreatment on the substrates contributed to the stability of the supported film.

As mentioned before, in this case, the as-spun films were not subjected to any further treatment. Nevertheless, we observe that the morphology on this film suggests a suitable phase segregation. At this point, we hypothesize that THF may have contributed to this effect due to the polar affinity between this solvent and the PLA phase [24].

To further confirm the change in the topography of the film, AFM measurements of both films were performed. Figure 6A displays an AFM image of the as-spun film, a 3D image, and a topographic profile of the film; the corresponding is also observed for the hydrolyzed film in Figure 6B. A change in the root mean square values (RMS) from 22.4 to 16.7 nm is indicative that the roughness of the film was smoother after the hydrolysis process. The 3D images of the films revealed a difference in the density of the pores. This was validated by 3D projection of the extracted morphology profile. Hence, after hydrolysis, the valleys of the pores become more defined (Figure 6B), whereas prior to hydrolysis, the valleys appear to be broader, suggesting that PLA domains are present at the interphase.

All in all, the hydrolysis of the as-spun films based on the synthesized PS-*b*-PLA copolymers may represent an attractive alternative to obtain porous films considering the rather simple preparation steps.

### 3.5. Binding Tests of Metal Salts Using Hydrolyzed Films

After PLA hydrolysis of the as-spun substrates, we decided to analyze the metal caption capacity of the obtained films. At this stage, we speculate that the generated porous thin films bear in their structure triazole groups available at the surface of the pores. These functional groups are capable of binding to metal cations through a coordination bond [37]. Hence, we proceeded to test the affinity of metallic moieties towards the porous PS film that could be interacting with the embedded triazole groups. The experiments were conducted by dipping the hydrolyzed film in a CuCl_2_ solution. We explored the metal caption with Cu^2+^ due to its affinity to imidazole and triazole derivatives [38].

After thoroughly rinsing and drying the used substrates were analyzed via SEM–energy dispersive X-ray spectroscopy (SEM-EDX), to survey the presence of Cu^2+^ moieties that may have remained attached to the porous matrix.

For this experiment, we used block copolymer **1A** (Table 1) subjected to the described spin coating and subsequent PLA hydrolysis procedures (Figure 5B). The utilized substrate for this experiment was pretreated with the silylation process to avoid delamination of the film as discussed above.

Figure 7A shows an SEM image of the film after being in contact with a CuCl_2_ solution. It is evident that, after a through rinsing, some large pieces of the metallic salt remain attached onto the surface of the film. Figure 7B displays the Cu mapping of the same substrate, Cu moieties on the film are clearly observed. However, a minimum portion of Cu salt is distributed on the film as it can be observed from the purple dotting pattern (Figure 7B). At this point, it is unclear whether these micro-moieties are coordinated to the triazole groups on the PS-film or whether the Cu salt is physically attached onto the surface.

Nevertheless, we could hypothesize that the PS-film possesses certain affinity towards the Cu salts as can be observed from the EDX images. A minimum quantity of the Cu salt is attached onto the film, possibly by the interaction between the triazole groups and the metallic center [10]. Future investigations will consider analyzing the effect of using an aromatic triazole junction instead of an aliphatic one (copolymer **1A**). This feature might improve the affinity of the porous film towards metallic centers.

### 3.6. PS-b-PLA Semi-Bulk and Bulk Samples as Templates for Porous Membranes

Self-assembly and phase segregation of block copolymers in thin films are governed by different variables to those applied to the preparation of bulk or semi-bulk samples. Complementary to the previous analysis, we carried out several experiments to prepare monoliths by using the synthesized PS_n_-*b*-PLA_m_ copolymers as starting materials. Following Method 1 described in the materials and methods section, a concentrated solution (*C* = 10.0% *w*/*w*) of copolymer **2A** in toluene was employed to execute these experiments. After complete dissolution of the copolymer, a highly viscous solution was obtained; afterwards, the solution was poured into an as-made Teflon mold (~60 mm^2^) at 50 °C. A slight increase in the temperature of the solution allowed the material to become more fluid when casting it into the mold. Once the solvent evaporated, the sample was annealed at 110 °C for 1 h and a thin copolymer film was recovered from the Teflon mold.

According to an SEM image of the material (Figure 8A), the exhibited morphology on the surface evidences the formation of some pores. The distribution of these pores is less homogenous than that observed on the wafer-supported thin films. This may suggest that it is possible to modulate experimental conditions to obtain more homogeneous patterns on the samples prepared in semi-bulk aiming to replicate the behavior observed on supported thin films. However, the thickness (0.11 mm) and brittleness of the isolated film made it difficult to handle and to characterize by SEM. The stability and mechanical properties are also important to avoid disintegration of the film during the subsequent hydrolysis step. Therefore, we re-attempted the experiment to increase the thickness of the final monolith. This was achieved by pouring the copolymer solution into a Teflon mold of smaller dimensions (~20 mm^2^). In addition, a higher temperature (160 °C) was employed in this experiment to further promote phase segregation. As a result, a thicker and more stable film was recovered from the mold (1.20 mm). Figure 8B shows a micrograph of the monolith surface with a more defined pore formation as compared to that in Figure 8A. Although the distribution of these pores may not be very homogeneous, this result suggests that a self-assembly process may occur as a result from the thermal annealing at semi-bulk conditions.

To continue with the established experimental scheme, a hydrolysis of the annealed sample (Figure 8A) was carried out as described in the Appendix A to form a PS-based porous matrix. Hydrolysis selectively removed PLA from the as-made monoliths by immersing the material in an excess of an alkaline solution to form a heterogeneous mixture. This reaction was monitored via ^1^H NMR, specifically, by observing the change in the integral value of the signal of –CH proton corresponding to the repetitive unit of PLA (‘E’ signal in Appendix A) in relation to the PS aromatic signal. Following the analysis of the ^1^H NMR spectra at different times, we determined that after 56 h, 97% of PLA had been removed from the thin film derived from copolymer **2A** (see Appendix A for details). A similar experiment was performed for the thicker sample shown in Figure 8B. In this case, a longer hydrolysis time was considered, after 77 h, only 74% of PLA was removed. Hence, a lower amount of PLA was removed when the bulk sample was thicker. This effect may be attributed to diffusional effects related to the fact that a larger amount of mass was confined into a smaller volume during the preparation of the thicker sample. Thus, PLA domains are more densely packed within the monolith. Nevertheless, it was possible to selectively remove and quantify the PLA phase from the bulk sample to form a PS porous matrix.

Furthermore, we decided to employ a second method that could enhance phase segregation in the bulk material and reveal the equilibrium morphology of the sample. Hence, by following Method 2 copolymer **2B** (Table 1) was placed inside a PEEK mold to trigger self-assembly of the phases in bulk. After annealing at 160 °C for 2 h, a monolith was recovered from the mold and characterized by SEM. The morphology of this sample did not exhibit the formation of pores as compared to previous examples. In this case, we observed that only a few circle-like structures were unevenly distributed on the surface and, as compared to previous examples, the size, shape, and distribution of these items were not similar to the expected morphology (see Appendix A for more details).

Although Method 1 exhibited better results for accessing the desired morphology, it is not conclusive that these conditions are suitable to ensure the self-assembly of the copolymer. It is important to further investigate the variables that rule dynamics of block copolymers in bulk to promote the formation of ordered structures at the mesoscale. In this manner, an improvement may be achieved by specifically evaluating the effects of such variables to obtain a functional porous matrix.

## 4. Conclusions

A series of PS_n_-*b*-PLA_m_ copolymers was synthesized by copper-catalyzed azide-alkyne cycloaddition (CuAAC) through homopolymer PS and PLA end functional groups. By ^1^H NMR analysis we determined the presence of triazole derivatives in the investigated block copolymers; these functional groups were located at the junction between PS and PLA blocks.

We were able to study the self-assembly behavior via SVA and TA of one of the synthesized copolymers by preparing thin films onto untreated and pretreated silicon wafer substrates. SVA of the film based on copolymer 1A, yielded a cylindric-like morphology with a mean diameter of ca. 125 nm after 45 min of vapor-annealing with o-xylene on an untreated substrate. The observed morphology is consistent with that of similar reports and the pattern presented an acceptable homogeneity as observed by SEM analysis. However, the incompatibility between the substrate and the film led to delamination of the film when attempting to hydrolyze the PLA. A sylilation pretreatment of the substrates was performed to avoid delamination of the films. As a result, this pretreatment allowed selective removal of the PLA without destruction of the film. Though the SVA of the pretreated substrates did not yield the expected morphology under the experimental conditions.

Based on the thermal analysis of the synthesized copolymers, we established the annealing temperatures to perform TA of copolymer 1A on the pretreated substrates. These experiments did not allow us to access the expected morphology at the investigated conditions (120 and 150 °C, 10 min).

Regardless, the segregation of the phases was evidenced on the morphology of the spin-coated films (as-spun samples), since homogenously distributed pore-like structures were observed without an annealing treatment. Subsequent hydrolysis of PLA on the as-spun films evidenced the formation of pores on the surface as proved by SEM and AFM. The obtained PS-porous film exhibited some affinity to metallic centers after being in contact with a Cu salt solution as revealed by SEM-EDX investigations. Triazole-functionalized porous PS is a suitable candidate to further study its metal-caption properties.

Semi-bulk and bulk experiments were performed to obtain monoliths based on the synthesized copolymers **2A** and **2B**. As evidenced by SEM, the thermally annealed samples prepared in semi-bulk exhibited an uneven distribution of pores under the working conditions (110 and 160 °C, 1 h). Further investigations of these copolymers in bulk need to address the variables that rule this process to ensure a self-assembly of the copolymer.

Overall, this contribution widens the fact that functional block copolymers remain as macromolecules with a vast potential due to the practically endless possibilities that arise from combining well-known RDRP processes and functionalization reactions. In addition, self-assembly studies of these polymers may be improved by varying the annealing and/or spin-coating variables when thin films are prepared by monitoring the morphology. Further investigations on this topic may also consider experiments to quantify the amount of metal bonded to the porous matrix.

## Data Availability

Not applicable.

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
