# Peer review of "Triazole-Functionalized Mesoporous Materials Based on Poly(styrene-block-lactic acid): A Morphology Study of Thin Films"

_polymers, 2022, doi:10.3390/polym14112231_

Round 1

Reviewer 1 Report

The manuscript entitled “Triazole-functionalized mesoporous materials based on poly(styrene-block-lactic acid): A morphology study of thin films” requires minor revision before publication. The author mainly described the synthesis of PS-b-PLA copolymer based on radical polymerization. The works in the morphology identification and structural observation is sound. Although the manuscript is ready to be accepted, several suggestions were made to authors for revision. Please find the comments below.

(1) The numbers in the reference section should be checked and meet the publication format of Polymers.

(2) What is the possible application of triazole-functionalized mesoporous pore? Also, please clarify the importance of aliphatic triazole derivative or aromatic triazole derivative if the idea could be strengthened in the section 3.5.

(3) At line 573 and 581, please note that there is no Figure 9.

(4) Compound ID 1A should be added in the caption through Figure 2~7 for clearance.

(5) In the conclusion section, the author mentioned that “ SVA of the untreated substrates yielded a cylindrical-like morphology with a mean diameter of 125 nm after 45 minutes of SVA using o-xylene as an annealing solvent”. The statement would be below to the synthesized Compound ID 1A or 2A according to the effect of selective solvent o-xylene and the PLA volume fraction. Please specifiy the meaning of the sentence.

Author Response

The manuscript entitled “Triazole-functionalized mesoporous materials based on poly(styrene-block-lactic acid): A morphology study of thin films” requires minor revision before publication. The author mainly described the synthesis of PS-b-PLA copolymer based on radical polymerization. The works in the morphology identification and structural observation is sound. Although the manuscript is ready to be accepted, several suggestions were made to authors for revision. Please find the comments below.

(1) The numbers in the reference section should be checked and meet the publication format of Polymers.

We thank the reviewer for the positive overview and comments. We have addressed this issue regarding the reference styles according to the format of Polymers.

(2) What is the possible application of triazole-functionalized mesoporous pore? Also, please clarify the importance of aliphatic triazole derivative or aromatic triazole derivative if the idea could be strengthened in the section 3.5.

As part of the introduction section, we mentioned several of the potential applications of the functionalized porous polymers. As suggested by reviewer number 3, we have added a couple of additional references to broaden the application range of this type of materials.

We also thank the reviewer for noticing the importance between aliphatic and aromatic triazole groups. Future research in this direction will consider this variable. Thus, we have further elaborated the discussion in section 3.5 to include this idea. Additionally, we have added ref. 10 to the following paragraph to strengthen this hypothesis (page 14, line 625 – 630).

“Nevertheless, we could hypothesize that the PS-film possesses certain affinity towards the Cu salts as can be observed from the EDX images. A minimum quantity of the Cu salt is attached onto the film, possibly by the interaction between the triazole groups and the metallic center.[10] Future investigations will consider analyzing the effect of using an aromatic triazole junction instead of an aliphatic one (copolymer 1A). This feature might improve the affinity of the porous film towards metallic centers.”

(3) At line 573 and 581, please note that there is no Figure 9.

We addressed this error by indicating the correct reference of the mentioned SEM image as Figure 8A on line 668 and Figure 8B on line 676 on page 15.

(4) Compound ID 1A should be added in the caption through Figure 2~7 for clearance.

We thank the reviewer for this observation. As suggested, we explicitly mentioned the corresponding compound 1A in each caption of Figures 2 to 7.

(5) In the conclusion section, the author mentioned that “ SVA of the untreated substrates yielded a cylindrical-like morphology with a mean diameter of 125 nm after 45 minutes of SVA using o-xylene as an annealing solvent”. The statement would be below to the synthesized Compound ID 1A or 2A according to the effect of selective solvent o-xylene and the PLA volume fraction. Please specifiy the meaning of the sentence.

We thank the reviewer for making this observation. We have explicitly indicated that copolymer 1A is only related to this statement. The corresponding sentence was amended to clarify this and was inserted in the Conclusions section as follows (page 16, lines 711 – 713):

SVA of the film based on copolymer 1A, yielded a cylindrical-like morphology with a mean diameter of ca. 125 nm after 45 min of vapor-annealing with o-xylene on an untreated substrate.”

Reviewer 2 Report

Comments:

This article reports the synthesis of poly(styrene-block-lactic acid) (PS-b-PLA) copolymers triazole rings as a junction between blocks by copper-catalyzed azide-alkyne cycloaddition (CuAAC). These copolymers can be used for the preparation of thin films and porous membranes onto untreated and pretreated silicon wafer substrates, which show morphology changes by self-assembly through solvent vapor annealing (SVA), thermal annealing (TA), and hydrolysis. And the PS-porous film exhibited some affinity to the Cu salt. In my opinion, although this paper contains interesting results, the amount of data in the article is insufficient and needs to be revised

Some problems/suggestions are listed below:

  1. The research background and significance are not mentioned in the abstract, only the working part is introduced.
  2. Foreign words need to be italicized and the abbreviation of the conclusion section does not mention the full name.
  3. Check the format of the reference, e. g., the sequence number appears twice and the 32nd citation is nonstandard.
  4. The authors discuss the morphology variety of the thin film on different substrates, but the difference between untreated and treated silicon wafers is not represented by characterization.
  5. The characterization of PS-b-PLA is too simple, characterization experiments such as FITR, and WCA should be provided.
  6. The sentences are too long, and the results and discussion sections do not need to describe the experimental procedure in detail.
  7. These articles can be cited, which is helpful to explore applications.

(Zhang Y, Li S, Xu Y, et al.  Nano Research, 2022: 1-13.  Song W, Zhang Y, Yu D G, et al. Biomacromolecules, 2020, 22(2): 732-742.)

Author Response

This article reports the synthesis of poly(styrene-block-lactic acid) (PS-b-PLA) copolymers triazole rings as a junction between blocks by copper-catalyzed azide-alkyne cycloaddition (CuAAC). These copolymers can be used for the preparation of thin films and porous membranes onto untreated and pretreated silicon wafer substrates, which show morphology changes by self-assembly through solvent vapor annealing (SVA), thermal annealing (TA), and hydrolysis. And the PS-porous film exhibited some affinity to the Cu salt. In my opinion, although this paper contains interesting results, the amount of data in the article is insufficient and needs to be revised

Some problems/suggestions are listed below:

  • The research background and significance are not mentioned in the abstract, only the working part is introduced.

We thank this reviewer for his/her kind observations and thorough review of our manuscript. We have revised these remarks and modified the abstract section accordingly.

  • Foreign words need to be italicized and the abbreviation of the conclusion section does not mention the full name.

We addressed these observations and formatted the foreign words accordingly. In the conclusions section, the utilized abbreviations were previously defined throughout the text. For better understanding, the sample ID’s were explicitly indicated as needed in this section.

  • Check the format of the reference, e. g., the sequence number appears twice and the 32nd citation is nonstandard.

The format of the references has been amended as well as that of reference 35 previously indicated as 32.

  • The authors discuss the morphology variety of the thin film on different substrates, but the difference between untreated and treated silicon wafers is not represented by characterization.

Figure 3A shows the morphology derived from an untreated substrate, whereas Figure 3B shows the morphology of a pretreated substrate under the stated experimental conditions. The pretreated substrate was indicated as 'sylilated' in the previous version of the manuscript; we have modified the caption of Figure 3 to clarify this issue.

  • The characterization of PS-b-PLA is too simple, characterization experiments such as FITR, and WCA should be provided.

Complementary characterization techniques such as FT-IR and/or WCA could provide valuable information on the structure of the copolymer and its behavior on the substrate, respectively. However, the scope of this research was set on the preparation and morphology of thin films from the synthesized block copolymers. At the stage of our research, it is quite complex to perform additional characterizations of the synthesized copolymers. For further synthetic and characterization analysis, the reader is referred to our previous publication indicated as reference 15 and other reports such as reference 30. Future investigations will definitively take into account this valuable suggestion to further understand the characteristics of these block copolymers using the suggested characterization techniques.

  • The sentences are too long, and the results and discussion sections do not need to describe the experimental procedure in detail.

We thank the reviewer for pointing out this observation. We have modified several sentences throughout the results and discussion section to reduce or delete the experimental details and avoid redundancies.

  • These articles can be cited, which is helpful to explore applications.

(Zhang Y, Li S, Xu Y, et al.  Nano Research, 2022: 1-13.  Song W, Zhang Y, Yu D G, et al. Biomacromolecules, 2020, 22(2): 732-742.)

We appreciate that the reviewer brought to our attention the mentioned publications. We have added such references (indicated as references 19 and 21) in the Introduction section to widen the range of applications of functionalized porous thin films.

Reviewer 3 Report

The manuscript (polymers-1727336) presents the synthesis of poly(styrene-block-lactic acid) copolymers with triazole rings as a connecting unit between blocks. A self-assembly process was observed for the obtained films while microphase segregation of the untreated films yielded 125 nm pore size. The affinity of PS was highlighted using SEM-EDX. The manuscript is logically arranged and well structured. I would recommend for publication with a couple of suggestions:

  1. The introduction part- novelty should state more clearly the advance of the synthesis method. What are the advantages of the method employed in this study?
  2. In the case of 60% conversions how it is achieved the separation of the homopolymers from the copolymers?
  3. Please elaborate the statement “an aromatic triazole unit may have a lower influence on the “ phase segregation of the copolymer compared to an aliphatic junction” by explaining the mechanism and introduction a reference at least.
  4. Scale bars should be added in Figure 3B, 4b, 5b, 8b
  5. The presence of the triazole on the surface should be confirmed by XPS performed on the surface

Author Response

The manuscript (polymers-1727336) presents the synthesis of poly(styrene-block-lactic acid) copolymers with triazole rings as a connecting unit between blocks. A self-assembly process was observed for the obtained films while microphase segregation of the untreated films yielded 125 nm pore size. The affinity of PS was highlighted using SEM-EDX. The manuscript is logically arranged and well structured. I would recommend for publication with a couple of suggestions:

The introduction part- novelty should state more clearly the advance of the synthesis method. What are the advantages of the method employed in this study?

As part of the introduction section, we attempted to highlight the combined use of RDRP and classic organic chemistry, such as click reactions. This enabled us obtaining specific block copolymers with azide groups as a junction between blocks at a determined stage of the utilized synthetic route. This synthetic strategy denotes a step forward in our research towards the preparation of functionalized porous materials. In this regard, we inserted the following sentence to address this question (page 2, line 106 – 108):

“In addition, the incorporation of “click” chemistry via cupper-catalyzed reactions allows targeting a specific molecule due to its stereo-selective mechanism.[13]”

  • In the case of 60% conversions how it is achieved the separation of the homopolymers from the copolymers?

For all copolymer samples the separation of the homopolymers precursors (i.e. PS and PLA) was executed by precipitation into a mixture of solvents (hexane:ethanol:methanol) using the described experimental conditions (see section 2.3). This experimental procedure allowed us isolating the copolymers as a precipitated solid. For instance, DOSY-NMR analysis confirmed the isolation of the respective pure block copolymer, as evidenced by the presence of a single diffusion coefficient signal (see SI, Figure S3).

  • Please elaborate the statement “an aromatic triazole unit may have a lower influence on the “ phase segregation of the copolymer compared to an aliphatic junction” by explaining the mechanism and introduction a reference at least.

We thank the reviewer for this keen observation. Indeed, this statement could be quite intricate to demonstrate at this stage of our investigation. Thus, we have restructured this observation in terms of the potential complexation effects of the aromatic or aliphatic triazole derivative. The following paragraph includes the mentioned modification (pages7 – 8, lines 352 – 358):

“Note that the synthesis of our block copolymers considered two derivatives to form the triazole groups, an aliphatic and an aromatic derivative, represented by compounds A and B in Table 1, respectively. It is expected that the presence of an aromatic or aliphatic triazole derivative unit may not largely affect the phase segregation of the copolymer. However, the presence of an aromatic derivative may contribute to the complexation with a metallic center on the porous matrix due to an electronic delocalization of the aromatic rings.[34]”

  • Scale bars should be added in Figure 3B, 4b, 5b, 8b

As suggested, we have added the corresponding scale bars to Figs. 3B, 4B, 5B, and 8B.

  • The presence of the triazole on the surface should be confirmed by XPS performed on the surface

We appreciate this suggestion of the reviewer. We agree that X-ray Photoelectron Spectroscopy (XPS) could help to determine the presence of nitrogen atoms contained in the triazole derivatives of the porous films. However, at this stage of our investigation, it is too complicated to accomplish this characterization. Future investigations in this direction will definitively consider measuring XPS of thin films obtained through a systematic set of experiments.

Round 2

Reviewer 2 Report

It has been revised based on reviews to comply with journal submission standards. It is recommended to accept.